# Development and evidence of validity of the HIV risk perception scale for young adults in a Hispanic-American context

**Patricio Mena-Chamorro**[☉]**, Rodrigo Ferrer-Urbina**[ID]**\***[☉]**, Geraldy Sepúlveda-Páez**[☉]**, Francisca Cortés-Mercado**[‡]**, Carolina Gutierrez-Mamani**[‡]**, Kyara Lagos-Maldonado**[‡]**, María Peña-Daldo**[‡]

Psychology and Philosophy School, University of Tarapacá, Arica, Chile

☉ These authors contributed equally to this work.
‡ These authors also contributed equally to this work.
* rferrer@academicos.uta.cl

## Abstract

HIV/AIDS is a public health problem that is transmitted through risky sexual behavior. The literature suggests that the perception of HIV risk is a motivator for the prevention of risky sexual behaviors. There is no culturally adapted scale to assess HIV risk perception in the Hispanic-American population. The aim of this research was to develop a scale to assess HIV risk perception in Hispanic-American young adults. A cross-sectional instrumental design was used, with a sample of students from the Chilean city with the highest HIV rates. Participants (n = 524) were between 18 and 33 years old, of whom 51% were women, 84.4% said they were heterosexual and 43.7% said they had not been tested for HIV/AIDS. The final scale has 9 items and 2 dimensions: (1) perceived risk susceptibility and (2) perceived risk severity. The results showed that the identified structure provided adequate levels of reliability ($\omega$ > .8) and presented evidence of validity, based on the internal structure of the test (i.e., using ESEM) and on the relationship with other variables (i.e., the sexual risk behaviors scale). In addition, the results showed strong invariance between the scores for men and women. It is concluded that the HIV risk perception scale has adequate psychometric properties to assess HIV risk perception in equivalent samples.

## Introduction

Acquired immunodeficiency syndrome (AIDS), caused by the human immunodeficiency virus (HIV), is a pandemic that affects thousands of people socially, physically and psychologically [1], which influences quality of life and well-being, increases the development of depression and anxiety [2–4].

These adverse effects acquire greater relevance when considering that 37.9 million people are currently living with HIV/AIDS, and in Latin America, 100,000 new cases were registered in 2018, with an estimated 1.9 million people currently living with HIV/AIDS on this continent.

**Data Availability Statement:** All relevant data are within the paper and its Supporting Information files.

**Funding:** R.F. acknowledges partial support from FONDECYT through grant FONDECYT INICIACIÓN N˚11170395 and from UNIVERSIDAD DE TARAPACÁ through grant PROYECTO DE INVESTIGACIÓN MAYOR, Nº 3738- 16.

**Competing interests:** The authors have declared that no competing interests exist.

In Chile, confirmed cases of HIV/AIDS have consistently increased, with a cumulative total of 43.386 HIV-positive cases since the first notification in 2017, with the regions of Arica y Parinacota, Metropolitana and Tarapacá having the highest prevalence [5]. The main affected age groups are those 20–24 and 25–29 years of age, which constitute 40.4% of the total number of confirmed cases in the country. There is also a large difference in number of people diagnosed with HIV by sex, with a ratio of 5.9 men to 1 woman [5, 6]. In Chile, in line with the world trend [7], this difference is even greater in men who have sex with men (MSM); this population has rates at least 20 times higher than those of the general population [8], which is mainly attributed to the predominance of anal sex, which has a higher risk of condom breakage and internal bleeding [9, 10].

Sexual risk behaviors (hereafter referred to as SRBs) are the main vector of HIV/AIDS transmission [11], and include 1) sexual activity with multiple partners, 2) inappropriate use of protective barriers (i.e., condoms), and 3) sexual activity under the influence of alcohol and/or drugs [12, 13]. There are several factors that influence SRBs, among which the risk perception of HIV is considered a central component for behavioral change in multiple health models [14–19], such as: a) the health belief model [20], which considers the expectations about future infection, as well as the valency assigned to it, as a motivational determinant for risk reduction [21], showing, for example, a statistically significant association between high expectations of HIV infection (together with negative valences) and increased condom use in undergraduate students [22]; b) protective motivation theory [23], which suggests that people assess risk based on beliefs about the perceived severity and probability of infection, adjusting their behavior as both dimensions are increased and noting, for example, that high-risk groups with more accurate beliefs about HIV attributed greater severity and perceived themselves to be more vulnerable, with greater motivation to use condoms [24]; c) the extended parallel process model [25], which indicates that when people are faced with a risk situation, two assessments are made, one on the severity and susceptibility of the threat and the other on the effectiveness of the response, so that the perception of risk would act together with self-efficacy levels for prevention [26], which has been evidenced in the joint power of interventions based on both self-efficacy and perceived risk susceptibility to increase condom use among college students [27]; and d) the AIDS risk reduction model [28], which states that SRBs reduction depends on three stages, the first of which is the recognition and labeling of behavior as risky, that is, the perceived susceptibility to risk [28], and noting that those who have increased their capacity to recognize themselves as subjects of risk (e.g., being diagnosed with a sexually transmitted infection [STI]), tend to decrease their risk behaviors [29].

Despite the importance of risk perception, much of the research continues to use single-item scales to measure the construct [30–34], which constitutes a limited ability to reflect the variability of people's perceptions [14, 35], and these single-item scales have less or no validity evidence to support them [36]. However, there are some scales to address risk perception that overcome these constraints; they include multiple items and have some evidence of reliability and validity. Some of them are specific [14], while others only include risk perception inside a set of other dimensions [37]. However, within the specific HIV risk perception scales, there is no consensus on their dimensionality [14, 30]. Included are scales assessing only the perceived probability of acquiring HIV at the present and future times [38]; two-dimensional scales (i.e., perceived HIV risk and perceived HIV vulnerability) [39] that consider perceived probability and potential perceived harm as two independent dimensions; and finally, in recent years, a three-dimensional approach (i.e., perceived risk of HIV scale) [40], which, in addition to perceived probability and the impact of potential consequences, incorporates "risk intuition," which is similar to perceived probability but with the difference that, instead of items being

oriented towards beliefs (i.e., "I think my chance of getting infected with HIV are. . ."), they are oriented towards affects (i.e., "I feel I am unlikely to get infected with HIV").

The aforementioned instruments, in general, have outdated psychometric evidence, and the central element underlying this study is that the available instruments have neither been developed in nor have cultural adaptations with evidence of reliability and validity suitable for use in Hispanic-American populations. Therefore, due to the need to obtain evidence of reliability and validity specific to each culture [36, 41], the purpose of this study was to develop a scale particularly for the Hispanic-American context.

For this purpose, we decided to develop a new scale, since the development of a scale from one's own culture has some advantages over translations and linguistic adjustments [42]; the aim was to offer a brief measurement instrument with full authorization for its free use and adaptation in sexual health studies.

The proposed scale covers most of the dimensions available in the scales designed to assess HIV risk perception based on two dimensions, since, in our review, we found that: 1) all the scales include a dimension that measures the perceived probability of acquiring HIV (called perceived susceptibility to HIV in this study), understood as a belief in the subjective possibility of experiencing an inherent threat to HIV risk, and 2) most scales and theoretical models include an assessment of future consequences; therefore we decided to incorporate a dimension called perceived severity of HIV, defined as the severity of the impact that people imagine living with HIV would have on their lives.

## Materials and methods

### Design and participants

This study had a cross-sectional and instrumental design [43].

A time-space sampling strategy, also known as time-location sampling, was used [44, 45]. The sample was made up of 524 young adults from the city of Arica who were between 18 and 33 years of age; there were 267 (51%) women, 254 (48.5%) men and 3 (0.5%) who did not respond. Overall, 84.4% (n = 442) said they were heterosexual, 81.3% (n = 426) said they had sex in the last 2 years and 43.7% (n = 229) said they had not taken an HIV/AIDS test. Demographic details are shown in Table 1.

Individuals who were 18 years of age or older (i.e., age of legal autonomy in Chile) and an active student of higher education were eligible for inclusion. No subsequent exclusion criteria were applied.

### Instruments

The HIV risk perception scale was an ad-hoc developed scale designed to assess (1) perceived susceptibility to HIV, defined as a belief in the subjective possibility of experiencing an inherent threat to HIV risk, and (2) perceived severity of HIV, defined as the magnitude of harm that would be caused if they were living with HIV (conditional aspect). The scales had differential instructions for each dimension. For perceived susceptibility, the instructions stated the following: "A series of statements related to HIV/AIDS are presented below. Please indicate to what extent you consider these statements to be true with respect to your own person, that is, if you consider that they are something that could happen to you." For perceived severity, the instructions stated: " Following, we request that you imagine being diagnosed with HIV/AIDS. Please indicate the degree to which you think the following domains of your life would be affected." The response options were four-point Likert behavioral/attitudinal statements, with differential references for perceived susceptibility to HIV (i.e., 0 = "False" to 3 = "True") and perceived severity of HIV (0 = "Not at all" to 3 = "Seriously").

**Table 1. Relevant demographic characteristics.**

| | | M (SD) or N (%) |
|---|---|---|
| Biological sex | Male | 267 (51.0%) |
| | Female | 254 (48.5%) |
| | Missing | 3 (0.5%) |
| Age (years) | | 22.74 (3.58) |
| Marital status | Single | 488 (93.1%) |
| | Married | 17 (3.2%) |
| | Civil union | 17 (3.2%) |
| | Missing | 2 (0.4%) |
| Sexual orientation | Heterosexual | 442 (84.4%) |
| | Homosexual | 33 (6.3%) |
| | Bisexual | 30 (5.7%) |
| | Missing | 19 (3.6%) |
| Number of sexual partners | | 5.30 (7.99) |
| Sexual activity in the last 2 years | Yes | 426 (81.3%) |
| | No | 83 (15.8%) |
| | Missing | 15 (2.9%) |
| Diagnosed with HIV/AIDS | Yes | 2 (0.4%) |
| | No | 520 (99.2%) |
| | Missing | 2 (0.4%) |
| In the last 2 years, they have used protective barrier methods | Yes, regularly | 319 (60.9%) |
| | Never | 195 (37.2%) |
| | Missing | 12 (2.1%) |
| HIV/AIDS tests performed | Yes, regularly | 248 (47.3%) |
| | Never | 269 (51.3%) |
| | Missing | 7 (1.0%) |
| HIV/AIDS tests requested from your sexual partner | Yes, regularly | 186 (35.5%) |
| | Never | 327 (62.4%) |
| | Missing | 11 (2.1%) |
| Diagnosed with STIs | Never | 497 (94.8%) |
| | Only once | 19 (3.6%) |
| | Twice | 1 (0.2%) |
| | More than one disease | 2 (0.4%) |
| | Missing | 5 (1.0%) |

M = Mean; SD = Standard deviation; N = Number of subjects; % = Percentage.

A total of 40 items (20 for perceived susceptibility to HIV and 20 for perceived severity of HIV) were initially written. The initial proposal was evaluated by three expert judges (i.e., three PhDs, one expert in psychometry and two in health) who individually scored each of the items in relation to their representativeness and grammatical suitability. The experts suggested keeping 30 items, with which an online pilot study was applied in a sample of university students (n = 215). Then, the scale was tested iteratively based on the analysis and reliability of the items (i.e., those items with values < .30 in the corrected homogeneity coefficient were eliminated in an attempt to obtain good internal consistency indicated by ω < .80 or α < .70 estimations) [46]. Finally, a version with 23 items was applied for this study. The final version (see S1 and S2 Protocols) and its psychometric evidence are reported in the results section.

The sexual risk behaviors scale [12] is a 12-item scale designed to assess three dimensions of sexual risk behaviors: (1) sexual activity with multiple partners (items = 4), (2) inadequate or insufficient use of protective barriers (items = 4), and (3) sexual activity under the influence of alcohol or drugs (items = 4). The response options correspond to behavioral/attitudinal statements on a Likert scale of 4 points (0 = "never" and 3 = "always"), which are conditioned on reporting only the behavior of the last 2 years. The scale reported evidence of validity based on internal structure and good reliability ($\omega > .8$) [12].

## Procedures

This research was approved by the Scientific Ethics Committee of the University of Tarapacá.

Eight fifth-year psychology students were trained to give instructions, answer participants' questions and administer the questionnaires in a pencil and paper format in the city of Arica between March 2018 and May 2018. Participants were contacted by surveyors in the recreational areas of the higher education institution (reading areas, indoor courtyards, library, etc.); the surveyors explained the objectives of the study and invited the students to freely respond on the spot, without payment of any kind. Informed consent was obtained from each subject, after the research objectives, participant rights, anonymity and confidentiality were established. Anonymity was safeguarded by the anonymous return of the questionnaire in a sealed envelope without any kind of personal identification data. The response procedure lasted less than 15 minutes.

## Statistical analysis

To establish evidence of validity based on the internal structure of the scale, an exploratory structural equation model (ESEM) with GEOMIN rotation [47] and the robust weighted least squares estimation method (WLSMV), which is robust with non-normal discrete variables [48], was performed (model M1). The analysis was made from the polychoric correlations matrix [49]. The general fit of the model was assessed following the cut-off point recommendations proposed by Schreiber (e.g., CFI>.95, TLI>.95, and RMSEA < .06) [50]. Later, to make a more brief and optimized scale, a revised version was established by, removing items on the basis of three criteria: (1) retention of strong factorial loadings ($\lambda > .5$) [51], (2) removal of redundant items [52], and (3) removal of items with strong cross-loadings (>.3) [53, 54]. Additionally, based on the debugged structure, four models were tested: models with two covariate factors using ESEM-GEOMIN (M2a) and CFA (M2b) approaches; one-factor CFA (M3); second-order CFA (M4) with first-order factor variance fixed to 1; and bi-factor CFA (M5).

Reliability was estimated for each dimension with Cronbach's alpha and McDonald's hierarchical omega coefficients, both in their non-ordinal versions [55]. To evaluate the stability of the scale between people of different sexes, metric and scalar invariance tests were performed, and decreases in CFI under .005 and increases in RMSEA under .010 were regarded as evidence of invariance [56]. Finally, evidence of validity based on the relationships with other variables was established from a structural equation model of the relationships between the scale dimensions and the dimensions of the sexual risk behavior scale [12] using the WLSMV estimation method and the polychoric correlations matrix. All analyses were performed using Jamovi (9.0) and Mplus (8.0).

## Results

Table 2 shows the fit statistics of the measurement models.

**Table 2. Global fit of the measurement models.**

| Model | N.° Par | $\chi^2$ | df | p | CFI | TLI | RMSEA | RMSEA 90% CI | |
|---|---|---|---|---|---|---|---|---|---|
| | | | | | | | | Low | Upp |
| M1 | 115 | 1033.16 | 208 | .000 | .949 | .958 | .087 | .082 | .092 |
| M2a | 44 | 42.687 | 19 | .001 | .996 | .993 | .049 | .029 | .068 |
| M2b | 37 | 52.185 | 26 | .002 | .996 | .995 | .044 | .026 | .061 |
| M3 | 36 | 1056.42 | 27 | .000 | .847 | .796 | .270 | .256 | .284 |
| M4 | 37 | 51.426 | 26 | .000 | .996 | .995 | .043 | .025 | .061 |
| M5 | 45 | 39.014 | 18 | .000 | .997 | .994 | .047 | .027 | .068 |

M1 = ESEM with two covariate factors, 23 items; M2a = ESEM with two covariate factors, 9 items; M2b = CFA with two covariate factors, 9 items; M3 = one-factor CFA, 9 items; M4 = second-order CFA,9 items (with first order factor variance fixed to 1); M5 = bi-factor CFA,9 items; N° Par = number of parameters; $\chi^2$ = chi square; df = degrees of freedom; p = significance; CFI = comparative adjustment index; TLI = Tucker-Lewis index; RMSEA = error of the mean square of the approximation root. CI = confidence interval; Low = lower; Upp = upper.

According to the most common fit criteria (CFI > .95, TLI > .95, and RMSEA < .06) [50], the original model (M1) was not enough of an explanation for the observed covariations matrix (for item descriptions, loadings and cross-loadings of the 23 items version) (S1 Table), but models based on the 9-items debugged scale, with the exception of M3, showed good fit for most of the standards (i.e., M2a, M2b, M4 and M5). However, in both the ESEM (M2a) and CFA (M2b) approaches, the two-factor covariate model appears to be the most parsimonious model, since M4 showed a second-order factor that explains most of the perceived susceptibility (β = .70) and very little of the perceived severity (β = .18), while the general factor of M5 model showed only medium loadings (λ = .29–.54) with perceived severity and mild or no perceived susceptibility (λ = .03–.32), thus not representing a general dimension that can be interpreted. For illustrative purposes, Fig 1A and 1B show the models based on the 9-items version (i.e., M2a, M2b, M3, M4 and M5) with standardized loadings.

Based on the two-factor ESEM model, the final debugged scale had 9 items, divided into two covariate dimensions (i.e., perceived susceptibility to HIV with 4 items and perceived severity of HIV with 5 items). Factor loadings, factor covariances and reliability estimates for each dimension are presented in Table 3.

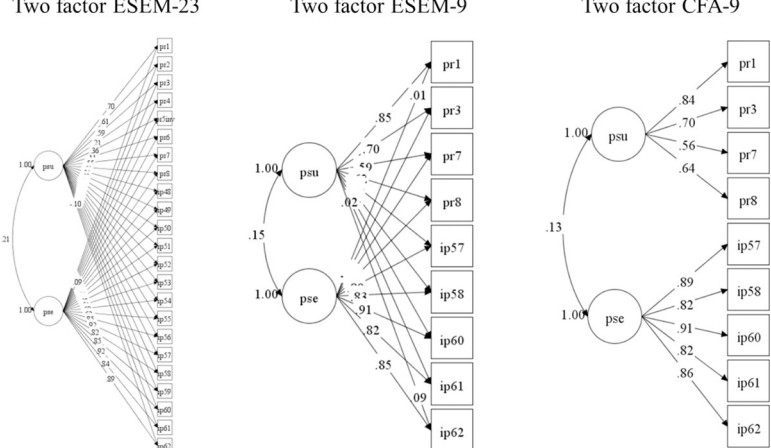

**Fig 1.** a. Graphical representation of the M1, M2a and M2b models. b. Graphical representation of the M3, M4 and M5 models.

**Table 3. Standardized factor loadings, factorial covariations and reliability coefficients (Cronbach's alpha and McDonald's hierarchical omega) for each dimension.**

| Original item (untested translation for comprehension purposes only) | PSu | PSe |
|---|---|---|
| Perceived susceptibility of HIV | | |
| Podría contraer VIH/SIDA como cualquier otra persona. (You could get HIV/AIDS just like anyone else) | **.853**** | .009 |
| Podría ser portador de VIH sin saberlo. (You could be HIV-positive without knowing it) | **.702**** | -.095 |
| Podría estar contagiado de VIH y no presentar síntomas. (You could be infected with HIV and have no symptoms) | **.591**** | -.042 |
| Me preocupa infectarme de VIH/SIDA. (I'm worried about getting HIV/AIDS) | **.617**** | .058 |
| Perceived severity of HIV | | |
| Mi desarrollo personal. (My personal development) | .028 | **.889**** |
| Mi vida laboral. (My working life) | .013 | **.827**** |
| Mi vida diaria. (My daily life) | -.040 | **.913**** |
| La relación con mis cercanos. (TMy close relationships) | -.062 | **.821**** |
| Mis expectativas y metas a largo plazo. (My expectations and long-term goals) | .069 | **.846**** |
| Factorial covariations | **.15**** | |
| Reliability estimators | PSu | PSe |
| Alpha (α) | .692 | .701 |
| Omega (ω) | .904 | .905 |

Factor loads >.4 are in bold

**p < .01; *p < .05. PSu = Perceived susceptibility to HIV; PSe = Perceived severity of HIV.

Factor loadings had strong representations of each factor (λ ≥ .50), without statistically significant cross-loading. Structural relationships between perceived susceptibility to HIV and perceived severity of HIV were low (r > .1) [51]. Reliability estimates were satisfactory (α > .80) [46] in the case of McDonald's hierarchical omega and sufficient (α > .70) [46] in the case of Cronbach's alpha.

Table 4 shows the results of the invariance tests of the final version of the scale (9-item version) between men and women. In the metric or scalar model compared to the configural model, the CFI and RMSEA deltas showed no practical changes of fit, with equivalence between factor loadings and factor intercepts, and therefore have the same meaning between the groups.

Finally, Table 5 and Fig 2 show the relationships between the latent dimensions of the developed scale and the SRB scale. The proposed model has acceptable fit (CFI = .958, TLI = .950, and RMSEA = .066).

According to the observed relationships, perceived susceptibility to HIV had a mild inverse effect (r < -.30) [51] on SAMP, a low direct effect (r > .10) [51] on IUPB, and a low inverse effect (r < -.20) [51] on SAIAD, with these effects being statistically significant (p < 0.05), while perceived severity of HIV had no significant effect on sexual risk behaviors.

## Discussion

The purpose of this study was to develop a scale to assess HIV risk perception for use in Hispanic-American young adults according to current psychometric standards. The fit statistics of the final model, for both the ESEM (M2a) and CFA (M2b) approaches; the sizes of the factorial loadings; and the non-existence of statistically significant cross-loadings, provide support for the two-dimensional model's structure, and evidence of validity was provided based on the internal structure for the adequate interpretation of the scores. In the same sense, estimates of

**Table 4. Measurement invariance testing.**

| | N.° par | $\chi^2$ | df | p | CFI | RMSEA | $\Delta_{\chi2}$ | $\Delta_{df}$ | $P_{\Delta\chi2}$ | $\Delta_{CFI}$ | $\Delta_{RMSEA}$ |
|---|---|---|---|---|---|---|---|---|---|---|---|
| Configural | 74 | 73.773 | 52 | .025 | .997 | .040 | – | – | – | – | – |
| Metric | 67 | 85.050 | 59 | .014 | .996 | .041 | 12.019 | 7 | .099 | -.001 | .001 |
| Scalar | 51 | 105.49 | 75 | .011 | .996 | .041 | 33.138 | 23 | .078 | -.001 | .001 |

N.° Par = number of parameters $\chi^2$: chi-square; df: degrees of freedom; p: significance; CFI = comparative adjustment index; RMSEA = error of the mean square of the approximation root; $\Delta_{\chi2}$: change in chi-square; $\Delta_{df}$: change in degrees of freedom; $\Delta p$: change in significance; $\Delta_{CFI}$: change in comparative adjustment index; $\Delta_{RMSEA}$: change in the error of the mean square of the approximation root.

the dimensional reliability coefficients allowed us to assume that each dimension had adequate internal consistency levels, which minimized measurement errors. Regarding the dimensionality of the scale, the comparison with alternative models (i.e., M3, M4 and M5) allowed us to suppose that the dimensions that constitute it were relatively independent aspects, since the covariation between the dimensions in the two-dimensional covariate model was low and none of the models that offered a general structure (M3, M4 and M5) presented an interpretable structure; therefore, both dimensions can be used as independent scales.

According to the invariance standards suggested by Chen [56], it is possible to support metric and scalar invariance between sexes (i.e., strong invariance); therefore, it is possible to apply the scale to both men and women because the factor loadings were equivalent between the groups and the dimensions had the same differential variability between sexes.

Regarding the evidence of validity based on the relationship with other variables, it was observed that the perceived susceptibility to HIV dimension partially explained the SRBs, as noted in previous research [15, 39, 40, 57–60]. The observed relationships between perceived susceptibility to HIV and SRBs were in the expected direction (i.e., protective factors), with the exception of the relationship with inappropriate use of protective barriers, which could be attributed to people who tend to not use condoms not perceiving themselves to be at higher risk of HIV infection. In the case of the perceived severity of HIV, no significant relationships were observed with SRBs, so use of this dimension to understand SRBs would be inappropriate. This background, accompanied by the low covariation between the dimensions of the scale (i.e., perceived susceptibility and severity), which leads to the assumption that they are independent dimensions (i.e., they can be used separately), would indicate that the perceived severity dimension does not constitute a relevant variable in the understanding of SRBs; however, it is possible that it has some practical relevance not detected in this study.

The main constraint of this study corresponds to the use of a non-probabilistic sampling strategy (i.e., a time-space method). Therefore, without data on the representativeness and generalizability of the population values estimated in this study, it is suggested to carry out new psychometric studies using this instrument in medical, health and educational contexts to increase the generalizability of the scale and obtain additional evidence of validity and

**Table 5. Correlation between the latent variables of the scales used in the study.**

| | SAMP | IUPB | SAIAD |
|---|---|---|---|
| Perceived susceptibility to HIV | -.345** | .187** | -.230** |
| Perceived severity of HIV | -.010 | -.052 | .025 |

$**p < .01$; $*p < .05$. SAMP = sexual activity with multiple partners; IUPB = inappropriate use of protective barriers; SAIAD = sexual activity under the influence of alcohol or drugs.

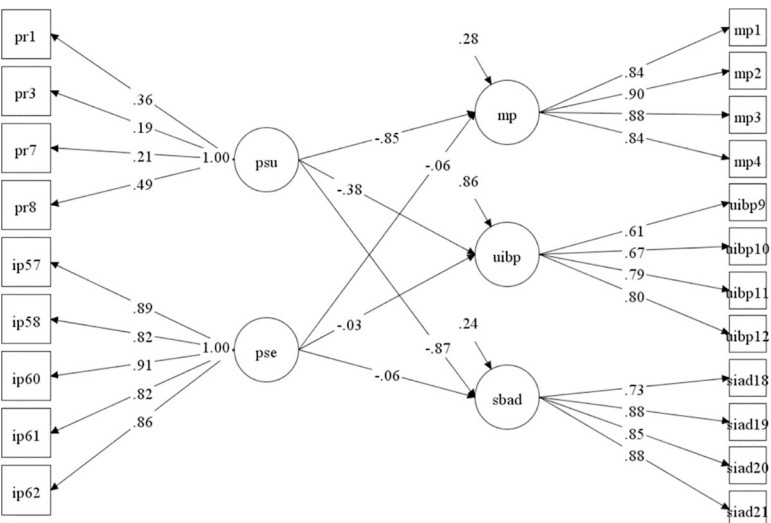

**Fig 2. Graphical representation of the SEM model.**

representativeness in other populations (e.g., high-risk populations, new countries, and migrant populations). Likewise, given the considerable differences in HIV rates between MSM compared to the rates for men and women who only have heterosexual sex, it is necessary to develop new invariance studies based on sexual orientation, since the small proportion of LGBTQ+ people in our sample did not allow for such tests.

The present scale is an instrument developed with current psychometric techniques that provides evidence of its function as a motivator towards prevention or involvement in SRBs [18, 19, 58]. The scale has a clear advantage of being a brief scale, that can be easily incorporated into a measurement battery for use in a health context, where quick tests are required conjointly with other scales. The evidence from this study suggests that the current scale can be used for the development of research on psychological factors involved in sexual behavior and related to HIV/AIDS prevention.

## Conclusions

The final 9-item version of the HIV risk perception scale has evidence of reliability, validity (i.e., based on the internal structure of the test and on convergence with other measures) and invariance of measures between sexes, which support the interpretation of scores in equivalent samples of men and women. The scale corresponds to independent dimensions, so it can be applied as a whole or as two different instruments.

## Supporting information

**S1 Protocol. Spanish version scale.**
(PDF)

**S2 Protocol. Unofficial English translated scale (translation for guidance only).**
(PDF)

**S1 Table. Item description, standardized factor loadings, standardized factor cross-loadings and factorial covariations for each dimension in 23 items version.**
(PDF)

**S1 Data. Raw data file (.Sav, SPSS file format).**
(RAR)

## Author Contributions

**Conceptualization:** Patricio Mena-Chamorro, Rodrigo Ferrer-Urbina, Geraldy Sepúlveda-Páez, Francisca Cortés-Mercado, Carolina Gutierrez-Mamani, Kyara Lagos-Maldonado, María Peña-Daldo.

**Data curation:** Francisca Cortés-Mercado, Carolina Gutierrez-Mamani, Kyara Lagos-Maldonado, María Peña-Daldo.

**Formal analysis:** Patricio Mena-Chamorro, Rodrigo Ferrer-Urbina, Geraldy Sepúlveda-Páez.

**Investigation:** Francisca Cortés-Mercado, Carolina Gutierrez-Mamani, Kyara Lagos-Maldonado, María Peña-Daldo.

**Methodology:** Patricio Mena-Chamorro, Rodrigo Ferrer-Urbina, Geraldy Sepúlveda-Páez.

**Writing – original draft:** Patricio Mena-Chamorro, Rodrigo Ferrer-Urbina, Geraldy Sepúlveda-Páez.

**Writing – review & editing:** Patricio Mena-Chamorro, Rodrigo Ferrer-Urbina, Geraldy Sepúlveda-Páez.

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
