## [Decision Letter · Decision Letter 0]

15 Jan 2020

PONE-D-19-32174

Development and evidence of validity of the HIV risk perception scale for young people and adults in a Latin American context.

PLOS ONE

Dear Dr. Ferrer,

Thank you for submitting your manuscript to PLOS ONE. After careful consideration, we feel that it has merit but does not fully meet PLOS ONE’s publication criteria as it currently stands. Therefore, we invite you to submit a revised version of the manuscript that addresses the points raised during the review process.

Dear authors,

We received two expert reviewers' feedback on your manuscript. As you can see by their comments below, there are both substantive and methodological questions that require clarification in the manuscript. 

We would appreciate receiving your revised manuscript by Feb 29 2020 11:59PM. To enhance the reproducibility of your results, we recommend that if applicable you deposit your laboratory protocols in protocols.io, where a protocol can be assigned its own identifier (DOI) such that it can be cited independently in the future. For instructions see: http://journals.plos.org/plosone/s/submission-guidelines#loc-laboratory-protocols

We look forward to receiving your revised manuscript.

Kind regards,

Jose A. Bauermeister, MPH, PhD

Academic Editor

PLOS ONE

Journal Requirements:

Reviewers' comments:

Reviewer's Responses to Questions

**Comments to the Author**

1. Is the manuscript technically sound, and do the data support the conclusions?

Reviewer #1: Partly

Reviewer #2: Yes

2. Has the statistical analysis been performed appropriately and rigorously? 

Reviewer #1: Yes

Reviewer #2: Yes

3. Have the authors made all data underlying the findings in their manuscript fully available?

Reviewer #1: Yes

Reviewer #2: Yes

4. Is the manuscript presented in an intelligible fashion and written in standard English?

Reviewer #1: Yes

Reviewer #2: No

5. Review Comments to the Author

Reviewer #1: This article develops and validates the HIV risk perception scale in Latin America. While the article has its strengths, there are several issues need to be addressed. The following are some of my suggestions:

1. Gay and bisexual men are the most disproportionately affected population by HIV in the United States. The study should have mentioned HIV risk differences between heterosexuals and homosexuals since men are more vulnerable to HIV than women in Chile. There is a possibility that high HIV risk among men in Chile is due to their sexual orientation.

2. If there is a significant difference in HIV risk between heterosexuals and homosexuals, the study needs to focus on invariance tests based on sexual orientation, not based on gender.

3. The author names different theories to support the importance of risk perception of HIV. However, the author did not show how the theory was applied in previous literature. It is recommended that the author explain how perceived susceptibility was acknowledged in each theory. In addition, detailed explanation (summary of results and study population) about each citation is needed (p 4, line 99)

4. According to the author, perceived susceptibility of HIV and perceived severity of HIV risk are predominant dimensions in the risk perception scales (p4, line 116). However, the author should have mentioned how each dimension was used and defined in the previous literature earlier in the article. There is a lack of why two dimensions were selected.

5. One of the study objectives is to examine HIV risk perception scale in the Latin American context. However, it is questionable why the study develops a new scale. The study could adapt existing scales and test adaptability of existing scales in the Latin American context. It was quite hard to find any Latin American context-specific item in the scale that was developed here.

6. The author needs to distinguish instruments and results. Results were included in the instruments section when it should have been carved out (p7, line 154-163).

7. Detailed information about perceived susceptibility of HIV and perceived severity of HIV risk is required in the instruments section. Although the author provides the actual items in Table 3, there is limited information in the prompt instructions to understand the items, such as “My personal development.”

8. Scale development is an exploratory study which means the researcher needs to provide theoretical support for each step as well as statistical support. However, the author did not mention theoretical meaning about deleting low factor loading items or making the debugged version. In addition, the study needs to provide support for statistical decision (i.e., OBLIMIN).

9. In the results section, the author describes two versions (the original version and the debugged version). However, the debugged version just pops up without any background information. Furthermore, the need for the debugged version is questionable. The author explains that the original model is insufficient to explicate the observed covariance matrix. However, the model fit was good enough to proceed as it is.

10. Figures are always helpful to understand the model. However, there were no figures in this study to help the reader understand model1 and model2. The author did not walk the reader through in the text either. It was quite hard to conceptualize the relationship between the latent variables.

11. In the beginning of the study, the author mentions dimensionality and selects two prominent dimensions. There are several ways to examine dimensionality, but the study did not fully examine the dimensionality. In order to define dimensionality, the author could compare the unidimensional, correlated factor, second-order, and bifactor models.

12. The author validates the developed scale by examining its relationship with SRB. However, the results showed that perceived severity was not related to SRB. Unexpected results in validity testing means the scale is not valid. The use of perceived severity scale is questionable unless the author give sufficient evidence based on previous literature. Also, the author could add more relevant variables to test validity.

13. Discussion is not thorough enough.

14. Some editing is needed to correct typos and grammatical errors.

Reviewer #2: This manuscript attempt to contribute to the development of culturally congruent measures. There is a significant need of data collection instruments in Spanish and of research conducted among populations at disproportionate risk for HIV infection in Latin America. Authors conducted a rigorous statistical analysis and propose a 9-item scale to measure HIV risk perception among young people and adults in Latin America.

While the intention of this work is well-aligned with pressing needs to address the HIV epidemic among Spanish-speaking populations, this manuscript fail to address crucial aspects in the development of culturally congruent measures. I am providing the following feedback with the goal of supporting and encouraging authors to revise this manuscript.

Overall, the manuscript will benefit from copy editing and revisions from an HIV scientist whose first language is English. The information included in the document is as expected for this kind of publication. However, the lack of clarity due to incorrect use of language is distracting and in instances misleading.

Authors should clarify and provide consistency in how to name the study sample. In the manuscripts they used terms such as Hispanic, Latino, Latin American, and Chilean to describe the study population. Some of these terms are not mutually inclusive.

Title: Current title might be misleading. Once authors decide on how to refer to the population they can also edit the title to be inclusive of all the population in the study (See comment below about age groups).

Materials and methods: More details about the study sample should be provided. For example, on Table 1 it is recommended to include the age range. Also, it seems like for certain characteristics included in the table not all participants provided an answer or some answers are missing as the n-size varies.

Regarding age groups – authors claimed (from the title on) that the scale is appropriate for young people and adults. However, they lacked to conduct analysis per age groups. Further, based on age M and SD, it seems like most participants were under 29. Pending on age distribution in the sample, authors could conduct further analysis and provide more specific recommendations for future research to test the use of this scale among young adults (<30yo). A measure sensitive of developmental characteristics would be an even stronger contribution to the field.

Instrument: Authors described that three expert judges assessed the items included in the scale. It is recommended to include the criteria used by the expert to keep 30 of 40 items. Was an expert panel approach used? If so, it should be described.

Discussion/Limitations: Authors limited the discussion of limitations to the sample size. There are other limitations related to statistical analyses that should be included in this section.

Table 3: It is recommended to include full items in Spanish as used in the scale. The items about “Perceived severity of HIV” are incomplete.

I also encourage authors to make findings from this study available to Spanish-speaking scientists. The Spanish-speaking community should also have access to these findings as they are the most likely to push this research work further.

6. PLOS authors have the option to publish the peer review history of their article (what does this mean?). If published, this will include your full peer review and any attached files.

Reviewer #1: No

Reviewer #2: No

---

## [Author Response · Author response to Decision Letter 0]

29 Jan 2020

Dear Editor:

First of all, I would like to thank you for the new chance given to this work, as well as the thorough and valuable guidance provided by both reviewers. The truth is that we have received the reviewers' contributions with much gratitude, since we consider that they have allowed us to improve substantially.

Given the value of the contributions received, we have tried to address each of the points noted and, in the few cases where we have not been able to do so, we have tried to explain them in detail. We will now proceed to provide the answers to each of the aspects noted (Comments are highlighted in red and responses in green):

Reviewer #1: 

C This article develops and validates the HIV risk perception scale in Latin America. While the article has its strengths, there are several issues need to be addressed. The following are some of my suggestions:

R Firstly, we wish to thank the reviewer 1 for the positive comments made and, mainly, for the relevant contributions made, which will undoubtedly mean a substantial improvement in the overall work. We have tried to incorporate most of the suggestions made and, on those points where we have not been able to, we have tried to answer the reasons for this decision.

C Gay and bisexual men are the most disproportionately affected population by HIV in the United States. The study should have mentioned HIV risk differences between heterosexuals and homosexuals since men are more vulnerable to HIV than women in Chile. There is a possibility that high HIV risk among men in Chile is due to their sexual orientation.

R Indeed, in Chile, as in most of the world, and as the reviewer has pointed out, men who have sex with men are the group with the highest prevalence of new HIV notifications, so we have proceeded to incorporate the following paragraph in lines 89-92: “In Chile, in line with the world trend [7], this difference is even greater in men who have sex with men (MSM), who have rates at least 20 times higher than general population [8], which is mainly attributed to the predominance of anal sex, which has a higher risk of condom breakage and internal bleeding [9,10]”

C If there is a significant difference in HIV risk between heterosexuals and homosexuals, the study needs to focus on invariance tests based on sexual orientation, not based on gender.

R Indeed, given the differences in risk levels between, for example, homosexuals and heterosexuals, it would be very interesting to be able to test invariance between these groups. However, given the small sample of people who identified themselves as homosexuals, it was impossible for us to carry out such an analysis and, given that in the psychological literature (Buss, 1995) sex is the variable that has shown to generate the greatest differences, we opted to test invariance between men and women. However, recognizing how interesting this analysis could be in subsequent studies, we proceeded to explicitly point it out as a limitation in the discussion between lines 330-333: “Likewise, given the considerable differences in HIV rates between MSM compared to men and women with only heterosexual sex, it is necessary to develop new invariance studies based on sexual orientation, since the small proportion of LGBTQ+ people in our sample did not allow for such test.”

Buss, D. M. (1995). Psychological sex differences: Origins through sexual selection.

C The author names different theories to support the importance of risk perception of HIV. However, the author did not show how the theory was applied in previous literature. It is recommended that the author explain how perceived susceptibility was acknowledged in each theory. In addition, detailed explanation (summary of results and study population) about each citation is needed (p 4, line 99)

R Indeed, we agree with the reviewer that the description of the above-mentioned health models is not enough, so we are grateful to the reviewer for allowing us to see the need to go deeper into them. Therefore, we incorporate the following paragraph in lines 96-117: “There are several factors that influence SRB, among which the risk perception of HIV stands out [14-19] being considered a central component for behavioral change in multiple health models, such as for instance: a) The health belief model [20], which considers the expectations about future infection, as well as the valency assigned to it, as a motivational determinant for risk reduction [21], showing, for example, a statistically significant association between high expectations of HIV infection, together with negative valences, with increased condom use, in undergraduate students [22]; b) protective motivation theory [23], which suggests that people assess risk based on beliefs about the perceived severity and probability of infection, adjusting their behavior as both dimensions are increased, noting, for example, that high-risk groups with more accurate beliefs about HIV attributed greater severity and perceived themselves to be more vulnerable, with greater motivation to use condoms [24]; c) extended parallel process model [25], which indicates that when people are faced with a risk situation, two assessments are made, one on the severity and susceptibility of the threat, and other on the effectiveness of the response, so that the perception of risk would act together with self-efficacy levels for prevention [26], which has been evidenced in the joint power of interventions based on both self-efficacy and perceived risk susceptibility to increase condom use among college students [27]; and d) AIDS risk reduction model [28], which states that SRB reduction depends on three stages, the first of which is the recognition and labeling of behavior as risky, that is, the perceived susceptibility to risk [28], noting that those who have increased their capacity to recognize themselves as subjects of risk (e.g. being diagnosed with an STI), tend to decrease their risk behaviors [29].” 

C According to the author, perceived susceptibility of HIV and perceived severity of HIV risk are predominant dimensions in the risk perception scales (p4, line 116). However, the author should have mentioned how each dimension was used and defined in the previous literature earlier in the article. There is a lack of why two dimensions were selected.

R Indeed, we agree with the reviewer that the description of the available scales and the selection criteria, for our proposal, were insufficiently substantiated. Therefore, we incorporated the detailed description of the scales and dimensions, available in the literature, between lines 124-134 " However, within the specific HIV risk perception scales, there is no consensus on their dimensionality [14, 30], with scales assessing only the perceived probability of acquiring HIV, at the present and future time [38]; two-dimensional scales (i.e. perceived HIV risk and perceived HIV vulnerability) [39] which consider, as independent dimensions, one that refers to perceived probability and another that refers to potential perceived harm; and finally, in recent years, a three-dimensional approach (i.e. Perceived Risk of HIV Scale) [40], which, in addition to perceived probability and the impact of potential consequences, incorporates "risk intuition", which is similar to perceived probability, but with the difference that items, instead of being oriented towards beliefs (i.e. "I think my chance of getting infected with HIV are"), are oriented towards affects (i.e. "I feel I am unlikely to get infected with HIV”) "; and we explained the reasons for our dimensional choice in lines 146-154: " The proposed scale covers most of the dimensions available in the scales designed to assess HIV risk perception, based on two dimensions, since, in our review, we found that: 1) all the scales include a dimension that measures the perceived probability of acquiring HIV, which we consider for the present development under the name of perceived susceptibility of HIV, understood as a belief in the subjective possibility of experiencing an inherent threat to HIV risk; and 2) given that most scales and theoretical models include an assessment of future consequences, we decided to incorporate a dimension called perceived severity of HIV, defined as the severity of the impact that people imagine on their lives, in the event of living with HIV.". 

C One of the study objectives is to examine HIV risk perception scale in the Latin American context. However, it is questionable why the study develops a new scale. The study could adapt existing scales and test adaptability of existing scales in the Latin American context. It was quite hard to find any Latin American context-specific item in the scale that was developed here.

R We understand the point made by the reviewer and are aware of the advantages of cultural adaptations, mainly the comparability between studies. However, in order for the adaptations to be truly comparable between them, they require the existence of invariance studies with the comparison versions, so unless this difficulty is overcome, the adaptation does not present any substantive advantage, with respect to local development. 

In this scenario, and given that the available scales require updates and/or do not have current psychometric analyses, we decided in favour of local development, in order to offer an intentionally brief scale, oriented to the development of correlational or screening studies, freely available and thought out from the local language. This last point does not necessarily imply that the items of the scales are particularly different from scales proposed in other languages, since the underlying concepts (i.e. perception and risk) are not far from other languages.

C The author needs to distinguish instruments and results. Results were included in the instruments section when it should have been carved out (p7, line 154-163).

R We understand the comment made by the reviewer, however, the mentioned results in the description of the scale, are intended to explain a previous process, with a pilot sample (i.e. not included in this or other studies), which we think it is needed to explain in order to facilitate the reader's understanding of the genesis of the version applied in this study. For this reason, we consider that the proposed writing structure can simplify the understanding and reading of the results, since, otherwise, we should describe previous analyses and samples, whose purpose was only a pilot, giving a density, which we believe to be unnecessary.

C Detailed information about perceived susceptibility of HIV and perceived severity of HIV risk is required in the instruments section. Although the author provides the actual items in Table 3, there is limited information in the prompt instructions to understand the items, such as “My personal development.”

R We thank the reviewer for noticing this and hope he will forgive our omission. All of the points made, have been corrected. 

The guidelines have been incorporated into the lines 173-179: “The scale has differential guidelines for each dimension: for perceived susceptibility, "A series of statements related to HIV/AIDS are presented below. Please indicate to what extent you consider these statements to be true with respect to your own person, that is, if you consider that it is something that could happen to you"; while, for perceived severity, " Following, we request that you imagine being diagnosed with HIV/AIDS. Please indicate the degree to which you think the following domains of your life would be affected"”.

C Scale development is an exploratory study which means the researcher needs to provide theoretical support for each step as well as statistical support. However, the author did not mention theoretical meaning about deleting low factor loading items or making the debugged version. In addition, the study needs to provide support for statistical decision (i.e., OBLIMIN).

R First of all, we apologize for an error in the rotation method used, which has already been corrected (OBLIMIN replaced by GEOMIN). Regarding the information on the criteria used, we extend the development in the following lines 213-224: “To establish the evidence of validity based on the internal structure of the scale, an exploratory structural equation model (ESEM), with GEOMIN rotation [47] and robust weighted least squares estimation method (WLSMV), which is robust with non-normal discrete variables [48], was performed (M1). The analysis was made from the polychoric correlations matrix [49]. The general fit of the model was evaluated following the cut point recommendation proposed by Schreiber (e.g. CFI>.95; TLI>.95; RMSEA<.06) [50]. Later, to make a briefer and optimized scale, a debugged version was established, through removing items, on the basis of 3 criteria: (1) retention of strong factorial loadings (λ >.5) [51]; (2) removal of redundant items [52] and (3) removal of items with strong cross-loadings (>.3) [53, 54]. Additionally, based on the debugged structure, four models were tested: two-factor ESEM with GEOMIN rotation (M2); one-factor CFA (M3); second-order CFA (M4); and bi-factor CFA (M5).”

47. Asparouhov T, Muthén B. Exploratory structural equation modeling. Structural Equation Modeling: A Multidisciplinary Journal, 2009; 16(3): 397-438. https://doi.org/10.1080/10705510903008204

48. Asparouhov T. Wald test of mean equality for potential latent class predictors in mixture modeling. Technical appendix. Los Angeles: Muthén & Muthén; 2007.

49. Barendse M, Oort F, Timmerman M. Using Exploratory Factor Analysis to Determine the Dimensionality of Discrete Responses. Structural Equation Modeling: A Multidisciplinary Journal, 2015; 22(1): 87-101. http://doi.org/10.1080/10705511.2014.934850.

50. Schreiber J. Update to core reporting practices in structural equation modeling. Research in Social and Administrative Pharmacy, 2017; 13(3): 634-643. https://doi.org/10.1016/j.sapharm.2016.06.006. 

51. Cohen J. Statistical power analysis for the behavioural sciences (2nd ed.). Hillsdale, NJ: Erlbaum; 1988.

52. Abad F, Olea J, Ponsoda V, García C. Medición en ciencias sociales y de la salud. Madrid: Síntesis, 2011. 26-38 p.

53. Muthén B, Asparouhov T. Bayesian structural equation modeling: A more flexible representation of substantive theory. Psychological Methods, 2012; 17(3): 313–335. https://doi.org/10.1037/a0026802

54. Xiao Y, Liu H, Hau K. A Comparison of CFA, ESEM, and BSEM in Test Structure Analysis. Structural Equation Modeling: A Multidisciplinary Journal, 2019; 26(5): 665-677. https://doi.org/10.1080/10705511.2018.1562928

C In the results section, the author describes two versions (the original version and the debugged version). However, the debugged version just pops up without any background information. Furthermore, the need for the debugged version is questionable. The author explains that the original model is insufficient to explicate the observed covariance matrix. However, the model fit was good enough to proceed as it is.

R We agree with the reviewer that the first model has a close to acceptable fit, although slightly below the standards we decided to use as a reference. However, fit was not the only criterion we used to conserve items and decide on the usefulness and interpretability of a model. We hope that the new description in the section on data analysis will be a satisfactory explanation of the choices made. However, to provide more transparency on the discarded items, an appendix C with items descriptions, loadings and cross-loadings of 23 items version.

C Figures are always helpful to understand the model. However, there were no figures in this study to help the reader understand model1 and model2. The author did not walk the reader through in the text either. It was quite hard to conceptualize the relationship between the latent variables.

R We agree with the reviewer and appreciate his suggestion, so we incorporate a comparative figure with all the models included in this study.

C In the beginning of the study, the author mentions dimensionality and selects two prominent dimensions. There are several ways to examine dimensionality, but the study did not fully examine the dimensionality. In order to define dimensionality, the author could compare the unidimensional, correlated factor, second-order, and bifactor models.

R We appreciate the important point made by the reviewer and proceeded to include all suggested models, as we agree that they are plausible models which are worth testing. However, the overall reported results did not differ, as the new tested models, although they have good fits, did not offer any interpretable structure.

C The author validates the developed scale by examining its relationship with SRB. However, the results showed that perceived severity was not related to SRB. Unexpected results in validity testing means the scale is not valid. The use of perceived severity scale is questionable unless the author give sufficient evidence based on previous literature. Also, the author could add more relevant variables to test validity.

R We agree with the reviewer that our results do not offer support for the usefulness of perceived severity for the study of sexual risk behavior, which detracts from evidence of validity for this dimension, as discussed in the article, in the lines 317-324: “In the case of the perceived severity of HIV, no significant relationships were observed with SRBs, so its use for understanding SRBs would be inappropriate. This background, accompanied by the low covariation between the dimensions of the scale (i.e. perceived susceptibility and severity), which leads to the assumption that they are independent dimensions (i.e. so they can be used separately), would indicate that the perceived severity dimension does not constitute a relevant variable in the understanding of SRBs, however, it is possible that it has some practical relevance not detected in this study”. Regarding the recommendation to use new external variables in order to obtain new evidence, unfortunately, in this application, we do not have other relevant variables, but this is undoubtedly something we will take into account in future work. 

C Discussion is not thorough enough.

R Indeed, we agreed with the reviewer that the development of the discussion was not enough, so we developed those points that we considered required more detail and incorporated some aspects that had not been addressed.

C Some editing is needed to correct typos and grammatical errors.

R We apologize for any mistakes made. We have made a second and third revision and hope that the grammatical and typographical inadequacies have been overcome. However, since we recognize our linguistic constraints, we make ourselves available to make corrections as often as necessary.

Reviewer #2:

C This manuscript attempt to contribute to the development of culturally congruent measures. There is a significant need of data collection instruments in Spanish and of research conducted among populations at disproportionate risk for HIV infection in Latin America. Authors conducted a rigorous statistical analysis and propose a 9-item scale to measure HIV risk perception among young people and adults in Latin America.

R We deeply appreciate the words of encouragement provided by reviewer 2, and are very pleased that she/he considers the study to be a potential contribution to the development of sexual health research in Spanish-speaking populations.

C While the intention of this work is well-aligned with pressing needs to address the HIV epidemic among Spanish-speaking populations, this manuscript fail to address crucial aspects in the development of culturally congruent measures. I am providing the following feedback with the goal of supporting and encouraging authors to revise this manuscript.

R We thank the reviewer for the comments made, which are undoubtedly a substantive contribution to the improvement of the manuscript.

C Overall, the manuscript will benefit from copy editing and revisions from an HIV scientist whose first language is English. The information included in the document is as expected for this kind of publication. However, the lack of clarity due to incorrect use of language is distracting and in instances misleading.

R We apologize for any mistakes made. We have made a second and third revision and hope that the grammatical and typographical inadequacies have been overcome. However, since we recognize our linguistic constraints, we make ourselves available to make corrections as often as necessary.

C Authors should clarify and provide consistency in how to name the study sample. In the manuscripts they used terms such as Hispanic, Latino, Latin American, and Chilean to describe the study population. Some of these terms are not mutually inclusive.

R We appreciate the comment made and apologize for these inconsistencies. Finally, we have chosen to refer to the Hispanic American population, since, given the simplicity and generality of the instrument, we believe that the wording of the items should not require linguistic adaptations. However, we recognize that the initial evidence of validity corresponds only to the Chilean population.

C Title: Current title might be misleading. Once authors decide on how to refer to the population they can also edit the title to be inclusive of all the population in the study (See comment below about age groups).

R Corrected.

C Materials and methods: More details about the study sample should be provided. For example, on Table 1 it is recommended to include the age range. Also, it seems like for certain characteristics included in the table not all participants provided an answer or some answers are missing as the n-size varies.

R We agree with the reviewer, so we incorporate the age range, in lines 159-160 “between 18 and 33 years of age”, and explicit the missing cases in the sample description table, in order to make sense of the discrepancies observed.

C Regarding age groups – authors claimed (from the title on) that the scale is appropriate for young people and adults. However, they lacked to conduct analysis per age groups. Further, based on age M and SD, it seems like most participants were under 29. Pending on age distribution in the sample, authors could conduct further analysis and provide more specific recommendations for future research to test the use of this scale among young adults (<30yo). A measure sensitive of developmental characteristics would be an even stronger contribution to the field.

R We agree with the comment on the need to consider possible effects of developmental stage, so it would be interesting to be able to carry out studies of invariance by age. However, unfortunately we are limited by the characteristics of the sample to test for such effects. 

However, thanks to the reviewer's comment, we realized that defining the scale for youth and adults is, at least, risky, so, since our population corresponds to people with the legal age, in Chile, but under 30, we considered and redefined our sample as young adults.

C Instrument: Authors described that three expert judges assessed the items included in the scale. It is recommended to include the criteria used by the expert to keep 30 of 40 items. Was an expert panel approach used? If so, it should be described.

R No panel expert was used. we changed our description, trying to add more clarity, in the lines 184-187: “The initial proposal was evaluated by three expert judges (i.e. three PhD, one expert in psychometry and two in health), who individually scored each of the items, in relation to their representativeness and grammatical suitability, suggesting to keep 30 items”.

C Discussion/Limitations: Authors limited the discussion of limitations to the sample size. There are other limitations related to statistical analyses that should be included in this section.

R We thank and agree with the reviewer. We add new limitations in lines 326-335: “The main constraint of this study corresponds to the use of a non-probabilistic sampling strategy (i.e. time-space) and, therefore, without data on the representativeness and generalization of the population values estimated in this study, so it is suggested to carry out new psychometric studies, using this instrument in medical, health and educational contexts, in order to increase the generalization of the scale, obtain additional evidence of validity and representativeness in other populations (e.g. high-risk populations, new countries, migrant population). Likewise, given the considerable differences in HIV rates between MSM compared to men and women with only heterosexual sex, it is necessary to develop new invariance studies based on sexual orientation, since the small proportion of LGBTQ+ people in our sample did not allow for such test.”.

However, regarding the mention of statistical limitations, we will gladly try to consider them if the reviewer provides us additional details.

C Table 3: It is recommended to include full items in Spanish as used in the scale. The items about “Perceived severity of HIV” are incomplete.

R We thank the reviewer for this comment and apologize for the omission. The point made has already been corrected.

C I also encourage authors to make findings from this study available to Spanish-speaking scientists. The Spanish-speaking community should also have access to these findings as they are the most likely to push this research work further.

R We agree with the reviewer and recognize the value of extending the opportunity for access to the scale and the study to Spanish-speaking populations. Therefore, if the manuscript is accepted, and if Plos One allows us to do so, we are committed to incorporating, as a complementary subject, a Spanish translation of the final version of the manuscript. For the time being, we have incorporated the Spanish application form, in Appendix A.

---

## [Decision Letter · Decision Letter 1]

18 Feb 2020

PONE-D-19-32174R1

Development and evidence of validity of the HIV risk perception scale for young adults in a Hispanic-American context.

PLOS ONE

Dear Dr. Ferrer,

Thank you for submitting your manuscript to PLOS ONE. After careful consideration, we feel that it has merit but does not fully meet PLOS ONE’s publication criteria as it currently stands. Therefore, we invite you to submit a revised version of the manuscript that addresses the points raised during the review process.

We would appreciate receiving your revised manuscript by Apr 03 2020 11:59PM. To enhance the reproducibility of your results, we recommend that if applicable you deposit your laboratory protocols in protocols.io, where a protocol can be assigned its own identifier (DOI) such that it can be cited independently in the future. For instructions see: http://journals.plos.org/plosone/s/submission-guidelines#loc-laboratory-protocols

We look forward to receiving your revised manuscript.

Kind regards,

Jose A. Bauermeister, MPH, PhD

Academic Editor

PLOS ONE

Reviewers' comments:

Reviewer's Responses to Questions

**Comments to the Author**

1. If the authors have adequately addressed your comments raised in a previous round of review and you feel that this manuscript is now acceptable for publication, you may indicate that here to bypass the “Comments to the Author” section, enter your conflict of interest statement in the “Confidential to Editor” section, and submit your "Accept" recommendation.

Reviewer #1: (No Response)

Reviewer #2: All comments have been addressed

2. Is the manuscript technically sound, and do the data support the conclusions?

Reviewer #1: Partly

Reviewer #2: Partly

3. Has the statistical analysis been performed appropriately and rigorously? 

Reviewer #1: N/A

Reviewer #2: Yes

4. Have the authors made all data underlying the findings in their manuscript fully available?

Reviewer #1: Yes

Reviewer #2: Yes

5. Is the manuscript presented in an intelligible fashion and written in standard English?

Reviewer #1: Yes

Reviewer #2: No

6. Review Comments to the Author

Reviewer #1: 1. The author presented a relevant theory and examples, but the way it was presented was hard to follow. Please reference other relevant work.

2. The author used ESEM to explore the factor structure. After the author confirmed the factor structure with 4 items in PSu and five items in PSe, the author could have compared three different CFA models – the correlated factor model (allowing correlation between latent PSu and PSe), the second-order, and the bifactor. The author’s rationale to select ESEM over other models was not developed enough (line 252-256).

3. The study assumed that there are two dimensions in the perception scale at the beginning. That is the reason why two dimensions were developed/asked with different prompts. However, the author used ESEM which assumes that each item can be explained by not only PSu, but also PSe. There is a discrepancy here. It looks like CFA fits better than exploratory study since the author already developed the item with two dimensions. Is there any reason for conducting ESEM?

4. It is nice that the study examined different models and presented in a figure. However, the author needs to walk readers through each model and its rationale with short descriptions.

5. The author conducted the second-order model with two latent variables but I wonder how it was even possible. A second-order factor needs at least three first-order factor indicators. However, in this study, there are only two first-order factors (PSu, PSe).

6. Poor quality of figure. In addition, the author could add another model related to Table 5.

7. More information is required to explain the debugged version. Also, the rationale for debugged version was not presented.

8. Overall, long sentences with lots of comma makes it hard for readers to follow along.

9. The author updated the discussion section, but it still needs more work. Discussion needs to include more synthesis and placing the findings in the context of the literature and how the findings contribute to the existing literature.

10. Editing is needed. (i.e., unclosed brackets, “Tabla”?)

Reviewer #2: Thanks for considering our recommendations and incorporating our feedback so diligently. Copy editing is recommended prior publication.

7. PLOS authors have the option to publish the peer review history of their article (what does this mean?). If published, this will include your full peer review and any attached files.

Reviewer #1: No

Reviewer #2: No

---

## [Author Response · Author response to Decision Letter 1]

11 Mar 2020

Dear Editor:

First of all, I would like to thank you for the new chance given to this work, as well as the thorough and valuable guidance provided by both reviewers. The truth is that we have received the reviewers' contributions with much gratitude, since we consider that they have allowed us to improve substantially.

Given the value of the contributions received, we have tried to address each of the points noted and, in the few cases where we have not been able to do so, we have tried to explain them in detail. Additionally, duo to our idiomatic constraints, we have submitted it to a proofreading service (https://www.proof-reading-service.com/, ref. no. 202003-50329) and adopted all suggested changes, so we hope to have increased the readability, significantly.

We will now proceed to provide the answers to each of the aspects noted (Comments are highlighted in red and responses in green):

First of all, we would like to deeply thank the exhaustive review carried out by the reviewer 1 and all the contributions made, which we recognize as extremely constructive and which have resulted in a notable improvement of the initial version.

Reviewer #1: 1. The author presented a relevant theory and examples, but the way it was presented was hard to follow. Please reference other relevant work.

Response: Due to our idiomatic constraints, we have sent the work to a proofreading service (https://www.proof-reading-service.com/, ref. no. 202003-50329) and adopted all suggested changes, so we hope to have increased the readability of this paragraph significantly. Regarding the suggestion to reference other relevant works, we ask the reviewer to please specify some relevant papers or authors that we have omitted in our review.

2. The author used ESEM to explore the factor structure. After the author confirmed the factor structure with 4 items in PSu and five items in PSe, the author could have compared three different CFA models – the correlated factor model (allowing correlation between latent PSu and PSe), the second-order, and the bifactor. The author’s rationale to select ESEM over other models was not developed enough (line 252-256).

Response: Although ESEM includes elements of exploratory factor analysis, and its name usually implies that this meaning is assigned to it, it is a confirmatory strategy, as are the CFA (Asparouhov, & Muthén, 2009; Marsh, Morin, Parker, & Kaur, 2014; Marsh et al., 2009), so our two-factor ESEM model is equivalent to the CFA model, with the only particularity that it recognizes the existence of cross-loadings. However, it seems to us that the reviewer's suggestion, to incorporate the CFA model, contributes to corroborate the strength of the structure, so we have included the equivalent CFA model. 

With regard to the rationality for the choice of the model , we have followed the reviewer suggestions and tried to express ourselves better in the following paragraph (lines 252-264): “According to the most common fit criteria (CFI > .95, TLI > .95, and RMSEA < .06) [50], the original model (M1) is not enough of an explanation for the observed covariations matrix (for items descriptions, loadings and cross-loadings of the 23 items version) (S3 Table), but models based on the 9 items debugged scale, with the exception of M3, show good fit standards in most of them (i.e., M2a, M2b, M4 and M5). However, both in the ESEM (M2a) and CFA (M2b) approaches the two-factor covariate model appears to be the most parsimonious model, since M4 shows a second-order factor that explains most of the perceived susceptibility (β = .70) and very few of the perceived severity (β = .18), while the general factor of M5 shows only medium loadings (λ = .29–.54) with perceived severity and mild or no perceived susceptibility (λ = .03–.32), thus not representing a general dimension that can be interpreted. For illustrative purposes, Figures 1a and 1b show the models based on the 9 items version (i.e., M2a, M2b, M3, M4 and M5) with standardized loadings.”

Asparouhov, T., & Muthén, B. (2009). Exploratory structural equation modeling. Structural equation modeling: a multidisciplinary journal, 16(3), 397-438. https://doi.org/10.1080/10705510903008204

Marsh, H. W., Morin, A. J., Parker, P. D., & Kaur, G. (2014). Exploratory structural equation modeling: An integration of the best features of exploratory and confirmatory factor analysis. Annual review of clinical psychology, 10, 85-110. https://doi.org/10.1146/annurev-clinpsy-032813-153700

Marsh, H. W., Muthén, B., Asparouhov, T., Lüdtke, O., Robitzsch, A., Morin, A. J., & Trautwein, U. (2009). Exploratory structural equation modeling, integrating CFA and EFA: Application to students' evaluations of university teaching. Structural equation modeling: A multidisciplinary journal, 16(3), 439-476. https://doi.org/10.1080/10705510903008220

3. The study assumed that there are two dimensions in the perception scale at the beginning. That is the reason why two dimensions were developed/asked with different prompts. However, the author used ESEM which assumes that each item can be explained by not only PSu, but also PSe. There is a discrepancy here. It looks like CFA fits better than exploratory study since the author already developed the item with two dimensions. Is there any reason for conducting ESEM?

Response: We understand the reviewer's concern and appreciate the comment. The reasons why we use ESEM is because, in recent years, it has become evident that most of the meassures present cross-loadings (Tóth-Király, Bõthe, Rigó, & Orosz, 2017) and the omission of these can cause trouble to identify the models (Xiao, Liu, & Hau, 2019). However, following the reviewer's suggestions, we have included the CFA model, in order to verify the stability of the proposed structure with two different techniques.

Xiao, Y., Liu, H., & Hau, K. T. (2019). A comparison of CFA, ESEM, and BSEM in test structure analysis. Structural Equation Modeling: A Multidisciplinary Journal, 26(5), 665-677.

Tóth-Király, I., Bõthe, B., Rigó, A., & Orosz, G. (2017). An illustration of the exploratory structural equation modeling (ESEM) framework on the passion scale. Frontiers in psychology, 8, 1968.

4. It is nice that the study examined different models and presented in a figure. However, the author needs to walk readers through each model and its rationale with short descriptions.

Response: We thank the reviewer for the comment, so we have provided a better explanation of the legend of each model and improved its graphic representations. 

5. The author conducted the second-order model with two latent variables but I wonder how it was even possible. A second-order factor needs at least three first-order factor indicators. However, in this study, there are only two first-order factors (PSu, PSe).

Response: We understand the reviewer's concern, since the estimation of a second-order model requires at least three latent variables. However, since the number of items and the Likert options of the scales of each latent variable are comparable, we have assumed that the second order variable has an equivalent weight in each first order factor, so we have set the latent variances to 1, which allows us to identify and estimate the model. To make this situation transparent, we have noted this (“second-order CFA (M4), with first order factor variance fixed to 1”) in statistical analysis (lines 227-228) and in the legend of table 2.

6. Poor quality of figure. In addition, the author could add another model related to Table 5.

Response: The figures were improved by dividing figure one into two (1a and 1b), and representations of all the models presented in table 2 were included. Additionally, we incorporated a figure 2 to represent the model in table 5.

7. More information is required to explain the debugged version. Also, the rationale for debugged version was not presented.

Response: We have tried to clarify our explanation, along the following lines:

(lines 222-225) “Later, to make a more brief and optimized scale, a debugged version was established by, removing items on the basis of three criteria: (1) retention of strong factorial loadings (λ > .5) [51], (2) removal of redundant items [52], and (3) removal of items with strong cross-loadings (>.3) [53, 54].”

(lines 252-264) “According to the most common fit criteria (CFI > .95, TLI > .95, and RMSEA < .06) [50], the original model (M1) is not enough of an explanation for the observed covariations matrix (for items descriptions, loadings and cross-loadings of the 23 items version) (S3 Table), but models based on the 9 items debugged scale, with the exception of M3, show good fit standards in most of them (i.e., M2a, M2b, M4 and M5). However, both in the ESEM (M2a) and CFA (M2b) approaches the two-factor covariate model appears to be the most parsimonious model, since M4 shows a second-order factor that explains most of the perceived susceptibility (β = .70) and very few of the perceived severity (β = .18), while the general factor of M5 shows only medium loadings (λ = .29–.54) with perceived severity and mild or no perceived susceptibility (λ = .03–.32), thus not representing a general dimension that can be interpreted. For illustrative purposes, Figures 1a and 1b show the models based on the 9 items version (i.e., M2a, M2b, M3, M4 and M5) with standardized loadings.”

8. Overall, long sentences with lots of comma makes it hard for readers to follow along.

Response: Due to our idiomatic constraints, we have sent the work to a proofreading service (https://www.proof-reading-service.com/, ref. no. 202003-50329) and adopted all suggested changes, so we hope to have increased the readability of this paragraph significantly.

9. The author updated the discussion section, but it still needs more work. Discussion needs to include more synthesis and placing the findings in the context of the literature and how the findings contribute to the existing literature.

Response: We have given a second reading to the discussion, adding some minor changes and trying to improve the synthesis. However, if the reviewer could give us more specific guidance, we would be pleased to address it.

10. Editing is needed. (i.e., unclosed brackets, “Tabla”?)

Response: Due to our idiomatic constraints, we have sent the work to a proofreading service (https://www.proof-reading-service.com/, ref. no. 202003-50329) and adopted all suggested changes, so we hope to have increased the readability of this paragraph significantly.

Reviewer #2: Thanks for considering our recommendations and incorporating our feedback so diligently. Copy editing is recommended prior publication.

Response: We are deeply grateful to the reviewer for his generous comments and for all the support for the improvement of the manuscript. We recognize our idiomatic constraints, so we have followed the reviewer's suggestion and submitted it to a proofreading service (https://www.proof-reading-service.com/, ref. no. 202003-50329) and adopted all suggested changes, so we hope to have increased the readability, significantly

---

## [Editor Report · Decision Letter 2]

16 Mar 2020

PONE-D-19-32174R2

Development and evidence of validity of the HIV risk perception scale for young adults in a Hispanic-American context.

PLOS ONE

Dear Dr. Ferrer,

Thank you for submitting your manuscript to PLOS ONE. After careful consideration, we feel that it has merit but does not fully meet PLOS ONE’s publication criteria as it currently stands. Therefore, we invite you to submit a revised version of the manuscript that addresses the points raised during the review process.

**Your submission still requires substantial editing for English grammar and usage. We ask that you please have the manuscript copyedited by either a native-English speaking colleague or a professional copy-editing service. While you may approach any qualified individual or any professional scientific editing service of your choice, PLOS has partnered with American Journal Experts (AJE) to provide discounted services to PLOS authors. AJE has extensive experience helping authors meet PLOS guidelines and can provide language editing, translation, manuscript formatting, and figure formatting to ensure your manuscript meets our submission guidelines. If the PLOS editorial team finds any language issues in text that AJE has edited, AJE will re-edit the text for free. To take advantage of this special partnership, use the following link: https://www.aje.com/go/plos/**.

We would appreciate receiving your revised manuscript by Apr 30 2020 11:59PM. To enhance the reproducibility of your results, we recommend that if applicable you deposit your laboratory protocols in protocols.io, where a protocol can be assigned its own identifier (DOI) such that it can be cited independently in the future. For instructions see: http://journals.plos.org/plosone/s/submission-guidelines#loc-laboratory-protocols

We look forward to receiving your revised manuscript.

Kind regards,

Jose A. Bauermeister, MPH, PhD

Academic Editor

PLOS ONE

Additional Editor Comments (if provided):

Your submission still requires substantial editing for English grammar and usage. We ask that you please have the manuscript copyedited by either a native-English speaking colleague or a professional copy-editing service. While you may approach any qualified individual or any professional scientific editing service of your choice, PLOS has partnered with American Journal Experts (AJE) to provide discounted services to PLOS authors. AJE has extensive experience helping authors meet PLOS guidelines and can provide language editing, translation, manuscript formatting, and figure formatting to ensure your manuscript meets our submission guidelines. If the PLOS editorial team finds any language issues in text that AJE has edited, AJE will re-edit the text for free. To take advantage of this special partnership, use the following link: https://www.aje.com/go/plos/.

---

## [Author Response · Author response to Decision Letter 2]

24 Mar 2020

Dear Editor:

Given our constraints with the English language, we have accepted your suggestion and sent our manuscript to the AJE editing service. We hope that these new improvements will allow us to obtain the desired standard for your prestigious journal. We attach the edition certificate.

Thank you.

---

## [Editor Report · Decision Letter 3]

26 Mar 2020

Development and evidence of validity of the HIV risk perception scale for young adults in a Hispanic-American context.

PONE-D-19-32174R3

Dear Dr. Ferrer,

We are pleased to inform you that your manuscript has been judged scientifically suitable for publication and will be formally accepted for publication once it complies with all outstanding technical requirements.

With kind regards,

Jose A. Bauermeister, MPH, PhD

Academic Editor

PLOS ONE

---

## [Editor Report · Acceptance letter]

6 Apr 2020

PONE-D-19-32174R3 

Development and evidence of validity of the HIV risk perception scale for young adults in a Hispanic-American context. 

Dear Dr. Ferrer:

I am pleased to inform you that your manuscript has been deemed suitable for publication in PLOS ONE. Congratulations! Your manuscript is now with our production department. 

With kind regards,

on behalf of

Dr. Jose A. Bauermeister 

Academic Editor

PLOS ONE